# Global Datasets of Leaf Photosynthetic Capacity for Ecological and Earth System Research

Jing M. Chen[1,2], Rong Wang[1], Yihong Liu[2], Liming He[3], Holly Croft[4], Xiangzhong Luo[5], Han Wang[6], Nicholas G. Smith[7], Trevor F. Keenan[8,9], I. Colin Prentice[10,6,11], Yongguang Zhang[12], Weimin Ju[12], and Ning Dong[10,11]

[1]School of Geography, Fujian Normal University,
[2] Department of Geography and Planning, University of Toronto,
[3]Canada Centre for Remote Sensing, Natural Resources Canada,
[4]School of Biosciences, University of Sheffield, Sheffield, UK,
[5]Department of Geography, National University of Singapore
[6]Department of Earth System Science, Tsinghua University, Beijing 100084, China,
[7]Department of Biological Sciences, Texas Tech University, Lubbock, TX, USA
[8]Climate and Ecosystem Sciences Division, Lawrence Berkeley National Laboratory, Berkeley, CA, USA
[9]Department of Environmental Science, Policy and Management, UC Berkeley, Berkeley, CA, USA
[10]Department of Life Sciences, Imperial College London, Silwood Park Campus, Buckhurst Road, Ascot SL5 7PY, UK
[11]Department of Biological Sciences, Macquarie University, North Ryde, NSW 2109, Australia
[12]International Institute of Earth System Science, Nanjing University, China

*Correspondence to*: Jing M. Chen (jing.chen@utoronto.ca)

**Abstract.** The maximum rate of Rubisco carboxylation ($V_{cmax}$) determines leaf photosynthetic capacity and is a key parameter for estimating the terrestrial carbon cycle, but its spatial information is lacking, hindering global ecological research. Here, we convert leaf chlorophyll content (LCC) retrieved from satellite data to $V_{cmax}$, based on plants' optimal distribution of nitrogen between light harvesting and carboxylation pathways. We also derive $V_{cmax}$ from satellite (GOME-2) observations of sun-induced chlorophyll fluorescence (SIF) as a proxy of leaf photosynthesis using a data assimilation technique. These two independent global $V_{cmax}$ products agree well ($r^2=0.79$, RMSE=15.46 $\mu$mol m$^{-2}$s$^{-1}$, P<0.001) and compare well with 3672 ground-based measurements ($r^2=0.69$, RMSE=13.8 $\mu$mol m$^{-2}$s$^{-1}$ and P<0.001 for SIF; $r^2=0.55$, RMSE=18.28 $\mu$mol m$^{-2}$s$^{-1}$ and P<0.001 for LCC). The LCC-derived $V_{cmax}$ product is also used to constrain the retrieval of $V_{cmax}$ from TROPOMI SIF data to produce an optimized $V_{cmax}$ product using both SIF and LCC information. The global distributions of these products are compatible with $V_{cmax}$ computed from an ecological optimality theory using meteorological variables, but importantly reveal additional information on the influence of land cover, irrigation, soil pH and leaf nitrogen on leaf photosynthetic capacity. These satellite-based approaches and spatial $V_{cmax}$ products are primed to play a major role in global ecosystem research. The three remote sensing $V_{cmax}$ products based on SIF, LCC and SIF+LCC are available at https://doi.org/10.5281/zenodo.6466968 (Chen et al., 2020) and the code for implementing the ecological optimality theory is available at https://github.com/SmithEcophysLab/optimal_vcmax_R (Smith, 2020).

# 1 Introduction

The Farquhar-von Caemmerer-Berry (FvCB) leaf photochemistry model (Farquhar et al., 1980) is widely used for simulating vegetation photosynthesis in ecological studies. The maximum carboxylation rate ($V_{cmax}$) that determines leaf photosynthetic capacity is an essential parameter in the FvCB model. The current state of the art (Rogers 2014, Rogers et al., 2017) in regional and global ecosystem modeling is to assign $V_{cmax}$ at 25˚C ($V_{cmax25}$) as a fixed parameter that varies by plant functional type (PFT), and is typically estimated from a ground-based database (Kattge et al., 2009 and 2020), even though observations show 2-3-fold variation in $V_{cmax25}$ for the same PFT. As the total simulated photosynthesis of a canopy is highly sensitive to $V_{cmax}$, this simple approach causes considerable distortion in modelled spatial distributions of the terrestrial carbon cycle (Bonan et al., 2011; Walker et al., 2017; Luo et al., 2017; Chen et al., 2019), hindering advancement in global ecological and Earth system research.

In recent studies, two independent satellite remote sensing approaches have been developed to estimate $V_{cmax}$ at the global scale. Since the first demonstration of sun-induced chlorophyll fluorescence (SIF) as a proxy of gross primary productivity (GPP) at the global scale (Frankenberg et al., 2011), the use of SIF for global carbon cycle estimation has been a highly active research field (Mohammed et al., 2019). He et al. (2019) attempted the first global mapping of $V_{cmax}$ from SIF after converting SIF observations into GPP that is related to $V_{cmax}$. A time series of daily $V_{cmax}$ maps was derived using SIF measured by the Global Ozone Monitoring Experiment-2 (GOME-2) sensor from 2007 to 2017 at 36 km resolution (see Methods). The second space-based approach to deriving $V_{cmax}$ is via leaf chlorophyll content (LCC). Chlorophyll harvests light that provides energy for the reactions in the Calvin–Benson–Bassham (CBB) cycle of photosynthesis, and therefore is likely coordinated with leaf carboxylation capacity ($V_{cmax}$) as plants optimize their photosynthetic nitrogen resources (Croft et al., 2017). The retrieval of LCC from satellite imagery offers the means of reliable and accurate LCC estimation over different spatiotemporal scales. Data from the MEdium Resolution Imaging Spectrometer (MERIS) in red, near infrared, and red-edge bands at 300 m resolution at 7-day intervals have been used to produce a global LCC map series from 2003 to 2012 (Croft et al., 2020). In a temperate broadleaf forest, it was found that LCC is better correlated with $V_{cmax}$ than leaf nitrogen content (LNC) over a growing season (Croft et al., 2017), and similar correlations between $V_{cmax}$ and LCC were established from empirical data for various PFTs (Luo et al., 2020; Lu et al., 2020). In this study, we use this new LCC time series with existing empirical LCC-$V_{cmax}$ relationshipes to derive another independent source of information for global $V_{cmax}$ assessment.

The $V_{cmax}$ products derived from SIF and LCC have different strengths and weaknesses. SIF contains strong signals for $V_{cmax}$ because it is directly related to the vegetation photosynthesis rate, but the spatial and temporal resolutions of existing satellite SIF observations are low. LCC can be derived reliably from multispectral satellite data at much higher spatial and temporal resolutions than those of SIF. Chlorophyll pigments have broad absorption features in the visible range and also affect the fine positioning of red-edge wavelengths. However, the derivation of LCC from remote sensing data is influenced by errors in

vegetation structural parameters used in the derivation. The conversion from LCC to $V_{cmax}$ depends on empirical relationships for different PFTs, which have considerable uncertainties (Luo et al., 2019). In order to make the best use of available satellite data for mapping $V_{cmax}$, we combined SIF and LCC data to produce a single global $V_{cmax}$ time series. We derived a global $V_{cmax}$ time series using SIF data from the TROPical Ozone Mission (TROPOMI) at 0.1˚ resolution in daily intervals for 2019 with LCC-derived $V_{cmax}$ as a constraint in the derivation using a parameter optimization technique (He et al., 2019; see also Methods). The constraint is made with LCC-derived $V_{cmax}$ aggregated to each 0.1˚ grid every 7 days as the initial value, which is then replaced when good quality TROPOMI SIF data are available. In this way the best information on $V_{cmax}$ from both SIF and LCC is combined. The combined global $V_{cmax}$ product is highly correlated with that produced from LCC ($r^2$=0.87, RMSE=12.04 $\mu$mol m$^{-2}$s$^{-1}$, P<0.001), suggesting that much of the LCC information is transferred to this product by filling in its data gaps.

The global distribution of $V_{cmax}$ has also been derived theoretically. Based on a new ecological optimality theory (Wang et al., 2017), Smith et al. (2019) calculated a global $V_{cmax}$ map from meteorological variables of radiation, air temperature and vapor pressure deficit using a monthly climate dataset (Harris et al., 2014). The theory proposes that leaves optimize the use of available resources so that the photosynthetic rate limited by $V_{cmax}$ equals that limited by the electron transport to generate ribulose-1,5,-bisphosphate (RuBP) needed in photosynthesis under average daytime conditions. In this theory, the electron transport rate is computed from meteorological conditions, and is independent of soil nutrient and water conditions. Evaluation against 3672 ground observations shows that the model can capture about 2/3 of the variance in the observed $V_{cmax}$ ($r^2$=0.66, RMSE=13.37 $\mu$mol m$^{-2}$s$^{-1}$, P<0.001), while the model bias is most significantly correlated to leaf nitrogen content among several leaf and soil parameters (Smith et al., 2019). The validity and reliability of $V_{cmax}$ information derived from the theory are yet to be evaluated outside of the limited amount of ground data.

Here we provide assessment of the reliability of these products for global ecological and Earth system studies. The specific objectives of this study are: (1) to derive new global $V_{cmax}$ products using satellite data; (2) to assess the accuracy of these products against a ground-based dataset; (3) to mutually assess these products; and (4) to evaluate the $V_{cmax}$ product derived ecological optimality theory using satellite-derived $V_{cmax}$ products, as the theory would be useful for estimating $V_{cmax}$ in prognostic terrestrial ecosystem models (TEMs) which are often used in Earth system models.

## 2. Methods

### 2.1. Deriving $V_{cmax}$ from SIF

During photosynthesis, plant leaves dissipate part of the excess light energy that is not used in photochemistry in the form of chlorophyll fluorescence (Porcar-Castell et al., 2014). Under conditions without strong moisture and/or thermal stress, the SIF emission from a leaf increases with its instantaneous photosynthetic rate (Frankenberg et al., 2011; Guanter et al., 2014; Sun

et al., 2014; Li et al., 2018; Wang et al., 2020), although SIF signals are small (1-5% of reflected radiation at near infrared wavelengths, Colombo et al., 2016) and contain noise from various sources including the variations in solar illumination angle and senor view angle (Dechant et al., 2020). In a plant canopy, sunlit leaves are the predominant sources of SIF (Pinto et al., 2016). For the purpose of deriving leaf-level information, the total SIF measured from a canopy was first separated into sunlit and shaded leaf components according to sun-target-sensor observation geometry and canopy architectural parameters. The

observation geometry was determined by satellite and solar zenith and azimuthal angles. The main canopy architectural parameters were leaf area index (LAI), which quantifies the amount of leaf area in the canopy per unit ground surface area, and the clumping index (CI), which characterizes the non-random spatial distribution of leaves in the canopy. Both LAI and CI were used to separate sunlit and shaded leaf fractions in the canopy, and the observation geometry determined the proportion of sunlit leaves observed by a satellite sensor (Chen et al., 1999). Leaf reflectance at the SIF wavelength was used to estimate

the strength of multiple scattering of emitted SIF in the canopy that enhances SIF observed from sunlit leaves (He et al., 2019). The sunlit SIF component derived in this way was then converted into the average sunlit leaf photosynthetic rate, from which $V_{cmax}$ is derived using a data assimilation technique (He et al., 2019). An ensemble Kalman filer (EnKF) was developed using an ecosystem model (Chen et al., 2012) and used in the data assimilation technique to optimize $V_{cmax}$ based on the difference between SIF-derived and modeled average sunlit leaf photosynthetic rates. In the optimization, it was assumed that the error

in modelling the photosynthetic rate was caused by both inaccuracy in the initial $V_{cmax}$ input (constants by PFT or estimated based on LCC) and the collective errors in other parameters including environmental conditions (meteorology and soil) used in the model. An error matrix was therefore developed to determine the amount of adjustment to the initial $V_{cmax}$ value (He et al., 2019). Optimized $V_{cmax}$ values often differed considerably from the initialized values beyond their error ranges, suggesting that SIF observations provided reliable and strong signals for its optimization, even though other model errors are also present.

The data assimilation methodology was first applied to GOME-2 SIF data and generated a global daily $V_{cmax}$ map series from 2007 to 2017 at 36 km resolution (He et al., 2019). In this study, this methodology was refined and applied to TROPOMI SIF data to produce global daily $V_{cmax}$ maps in 2019 at 0.1˚ resolution (approximately 10 km). The refinements included the conversion from SIF to GPP using non-linear relationships (Liu et al., 2022) rather than linear relationships used in He et al.

(2019) and the initialization of $V_{cmax}$ using the LCC product (Croft et al., 2020) rather than constant $V_{cmax}$ by PFT. Although the $V_{cmax}$ map series produced using TROPOMI SIF + LCC data is available for only one year, it has a much higher spatial resolution than that produced from GOME-2, and therefore has broader applications in global ecological research.

## 2.2. Deriving $V_{cmax}$ from LCC

LCC is responsible for light harvesting and providing excitation energy to drive photosynthesis in leaves, while $V_{cmax}$ defines the capacity of leaves to utilize the excitation energy for photosynthesis. These two leaf traits are dynamically optimized to local environmental conditions to achieve an optimal use of nitrogen resources (Xu et al., 2012). LCC is a relatively stable trait without much day-to-day and diurnal variations, while $V_{cmax}$ is sensitive to temperature. Empirical data show close

relationships between LCC and $V_{cmax25}$ (Houburg et al., 2013; Croft et al., 2017; Lu et a., 2020), which is $V_{cmax}$ normalized to
its value at 25˚C using a temperature function (Smith et al., 2019, see also Section 2.4 below). A two-step radiative transfer
model inversion method was developed for retrieving LCC from multispectral satellite data (Zhang et al., 2008; Croft et al.,
2020). In step 1, the canopy-level reflectance was inverted to leaf-level reflectance with a look-up-table (LUT) constructed
using canopy radiative transfer model for canopies with turbid media (Verhoef, 1984) and a geometrical optical model for
clumped canopies (Chen and Leblanc, 1997 and 2001) that computed observed sunlit leaf fraction according to canopy
structure and sun-target-view geometry. In step 2, the leaf-level PROSPECT model (Feret et al., 2008) was inverted to obtain
LCC from the inverted multi-spectral leaf reflectance. This two-step model inversion algorithm avoids issues with empirical
methods that directly link LCC to canopy-level remote sensing data, which lack generality because of variable canopy structure
and sun-target-view geometry. The first time-series of global LCC maps were retrieved using MERIS data from 2003 to 2011
at 300 m resolution and 7-day intervals (Croft et al., 2020). A validation using 248 ground sites in 5 PFTs suggests that this
product is reliable ($r^2$=0.5, p<0.01, RMSE=10.79 µg cm$^{-2}$ or mean error 23%). Using empirical relationships between LCC
and $V_{cmax25}$ for various PFTs (Luo et al., 2019), this global LCC time series was converted into $V_{cmax25}$.

### 2.3. Ground-based $V_{cmax}$ dataset

In this study, we use the same ground-based $V_{cmax}$ dataset as that used by Smith et al. (2019). It consisted of 3672 entries for
1474 plant species that are grouped into 7 PFTs in this study. Each entry consisted of $V_{cmax}$ measured from one or more leaves
with companion data of air temperature, humidity, incoming PAR, longitude and latitude. $V_{cmax}$ was derived from several pairs
of photosynthesis (A) and intercellular $CO_2$ concentration ($C_i$) (to construct an A/$C_i$ curve) (56%) or from a single pair of A
and $C_i$ using the method of De Kauwe et al. (2016) (44%).

To match with $V_{cmax}$ maps derived from SIF, LCC and EOT, the ground-based data were aggregated in two ways: (1) the data
points of the same PFT as that in the PFT map (Figure 4b) used for LCC and SIF processing were grouped to form an average
$V_{cmax}$ for a grid, while mismatched datapoints within the grid were ignored, and (2) all existing data points within each grid
were averaged and labelled as the dominant PFT. We found that second way resulted in better correlation with all three types
of $V_{cmax}$ maps. After the aggregation to 0.5˚ resolution, there were 180 data points for all PFTs used in the final analysis of all
$V_{cmax}$ products.

### 2.4. Temperature normalization

For the same leaf, $V_{cmax}$ varies exponentially with leaf temperature, and hence it is more meaningful to compare $V_{cmax25}$ between
leaves or different estimates. In this study, all global $V_{cmax}$ products are derived at the growth temperature. To facilitate their
comparisons with ground databases and their future use in models, $V_{cmax}$ is converted to $V_{cmax25}$ using a common temperature
function (Eq. 22 in Smith et al., 2019).

## 3. Results

### 3.1. Evaluation of Four Global $V_{cmax}$ Products

The global distributions of the growing season mean $V_{cmax}$ obtained from GOME-2 SIF, MERIS LCC and TROPOMI SIF are
shown in Figure 1 in comparison with $V_{cmax}$ calculated from the ecological optimality theory (EOT) at the growing season mean temperature. For this comparison, $V_{cmax25}$ derived from LCC is converted to $V_{cmax}$ at the mean growing temperature. The growing season is defined as the period when monthly mean air temperature is above 0°C. These four global $V_{cmax}$ distributions derived at different spatial resolutions at 36 km, 300 m, 0.1˚ and 0.5˚ from GOME-2 SIF, LCC, TROPOMI SIF and EOT, respectively, and aggregated to 0.5˚ resolution in Figure 1, are highly correlated spatially, although their details differ to some
extent. The distribution derived from the optimality theory appears to be spatially smooth, reflecting the fact that meteorological variables used for $V_{cmax}$ prediction do not vary abruptly in space. The three remote sensing products show mutually-consistent patchy patterns, suggesting that they have all captured some realistic variability on the ground associated with PFT distribution patterns. However, all four products show remarkable similarities in the overall geographic patterns and mutually well correlated with each other ($R^2$=0.76-0.90, p<0.001). Among three remote sensing products, SIF-derived products
correlate best with the product based on the ecological optimality theory (EOT) ($r^2$=0.85, RMSE=11.69 $\mu$mol m$^{-2}$s$^{-1}$, P<0.001 for GOME-2; $r^2$=0.76, RMSE=15.77 $\mu$mol m$^{-2}$s$^{-1}$, P<0.001 for TROPOMI). We further evaluate these products below.

All four $V_{cmax}$ products compare well with ground-based measurements (Figure 2) after they are aggregated to the corresponding 0.5° grids (see Methods). The correlation of optimality-based $V_{cmax}$ with the ground measurements is similar to
that shown in Smith et al. (2019) ($r^2$=0.66, RMSE=13.37 $\mu$ mol m$^{-2}$s$^{-1}$, P<0.001), and correlations of other three $V_{cmax}$ products with the same ground measurements are similar ($r^2$=0.69, RMSE=13.80 $\mu$ mol m$^{-2}$s$^{-1}$, P<0.001 for GOME-2; $r^2$=0.80, RMSE=8.99 $\mu$ mol m$^{-2}$s$^{-1}$, P<0.001 for TROPOMI; $r^2$=0.55, RMSE=18.28 $\mu$ mol m$^{-2}$s$^{-1}$, P<0.001 for LCC). It is encouraging to see that three of the four products captured about 2/3 of the variance in the ground data, despite the large-scale mismatch between the grids of these products and the ground data points. Some errors would also be expected from temporal mismatches
as the differences in the years of ground and remote sensing data acquisitions are not considered (in order to have as many data points as possible in the comparisons), although data outside of the growing season are excluded. While $V_{cmax}$ for individual leaves may vary greatly among plant species within the same functional type and with environmental conditions over the landscape, their locally averaged values would be expected to display a consistent spatial pattern at large scales that are determined more or less by meteorological conditions – permitting the success of the optimality theory for predicting $V_{cmax}$
based on meteorological variables alone. Coarse-resolution remote sensing data, such as GOME-2 SIF data at 36 km resolution and TROPOMI at 0.1° resolution, can also capture the spatial variability in $V_{cmax}$ at large scales.

The correlation statistics of the four products shown in Figure 2 with the ground database by plant function type are given in Table 1. Forest PFTs of ENF, DNF and DBF are combined in order to have a sufficient number of data points for the statistical analysis. Correlations for most PFTs are highly significant for all products (p<0.001), but for forest PFTs and SHR the correlations are weak for most products except for TROPOMI. The TROPOMI $V_{cmax}$ product (https://doi.org/10.5281/zenodo.6466968) compares best with the ground database in terms of the Pearson correlation coefficient ($r^2$), RMSE and the p value from two-tailed paired T tests, suggesting that the combination of SIF and LCC information is effective in capturing the spatial variability of $V_{cmax}$ for the various PFTs. It is therefore most ready for global ecological studies among the four products.

The $V_{cmax}$ values derived independently from GOME-2 SIF and MERIS LCC at the mean growing season temperature are well correlated overall ($r^2=0.79$) and for individual PFTs ($r^2=0.77-0.88$) except for EBF ($r^2=0.26$) (Figure 3). These high correlations suggest that the signals contained in both SIF and multi-spectral reflectance data used for LCC retrieval are strong and useful for deriving $V_{cmax}$. This is particularly encouraging because both types of remote sensing data are increasingly available with existing and forthcoming satellite sensors providing improved SIF (Muhammed et al., 2019) and multi-spectral data (such as the Sentinel sensor series, https://sentinel.esa.int/web/). The differences between these two independent retrievals of $V_{cmax}$ are still considerable, especially for EBF in the tropics due to frequent cloud cover, and there is room for improvement not only in the retrieval algorithms but also in providing improved SIF and spectral data at higher spatial and temporal resolutions. Much more ground-based data of $V_{cmax}$, LCC and associated structural parameters (leaf area index and clumping index) are still needed for refining and validating the retrieved $V_{cmax}$. However, these existing products are already a large step forward from the current state of the art and can be employed immediately for parameterizing and benchmarking TEMs. In other words, these products may have already overcome the $V_{cmax}$ bottleneck in accurate modeling of the spatio-temporal patterns of the terrestrial productivity and carbon cycle.

### 3.2. Influence of Environmental Factors on $V_{cmax}$

$V_{cmax}$ values derived from all three remote sensing products shown in Figure 1 are most obviously larger than those produced by EOT over croplands and grasslands in Americas, India and China. Cropland and grassland management, including fertilization and irrigation, may explain part of this divergence. To explore the possible influences of cropland and grassland irrigation on $V_{cmax}$, we used a global irrigation map (http://www.fao/nr/water/aquastat/irrigationmap/index.stm) at 0.5˚ resolution to compare with the relative difference ($\Delta V_{cmax}$) between TROPOMI SIF+LCC $V_{cmax}$ and EOT-based $V_{cmax}$, i.e. (TROPOMI-EOT)/EOT (Figure 4). Globally, $\Delta V_{cmax}$ increases with irrigated area percentage. The regression coefficient (R) between the actual values of percent irrigated area and $\Delta V_{cmax}$ for the areas of irrigation greater than 5% at the global scale is +0.32 (p<0.01) and +0.3 (p<0.01) for croplands and grasslands, respectively. In some regions, including China, India and the Middle East, the correlation coefficient is considerably higher (0.25-0.5, p<0.001). Irrigation in cropland and grassland would

reduce water stress and increase leaf photosynthetic capacity (Reed and Loik, 2016; Chen et al., 2019; Song et al., 2021), giving rise to the positive correlation between $\Delta V_{cmax}$ and percent of irrigation area in a pixel. For crops, fertilization would generally co-occur with irrigation (Sela, 2021), so this positive correlation could also include the effect of fertilization on Vcmax.


Soil properties may also influence $V_{cmax}$ (Reich et al., 2007, Maire et al., 2015; Ali et al., 2015, Smith and Dukes, 2018). Among soil properties available in the global SoilGrids database (Hengl et al., 2017), we found that soil pH best correlates with $\Delta V_{cmax}$. Soil pH is spatially variable (Figure 5a), and we found that $\Delta V_{cmax}$ is positively and significantly correlated to soil pH in most regions, with 40.3% (11163 out of 27681 pixels) of cropland and grassland areas having R>0.1 and p<0.1 (Figure

5). Similar statistics are found for other PFTs, suggesting that soil pH has similar effects on $V_{cmax}$ across PFTs. However, $\Delta V_{cmax}$ is not significantly correlated with other soil properties, including soil carbon content, nitrogen content and cation exchange capacity. Soil pH is a key control on soil biochemical reactions affecting nutrient uptake (Hall et al., 1998) and has an optimum range for plant growth from 5.5 to 7.5 (Islam 1980). As soil pH varies in a wide range (4 to 8, Figure 5), its effect on $V_{cmax}$ is therefore detectable from remote sensing signals. Plants on acidic soils tend to have higher ratios of leaf-internal to

ambient $CO_2$ (Wang et al., 2017; Dong et al., 2020; Pailassa et al. 2020) and therefore would be expected to have lower $V_{cmax}$. These results suggest that $V_{cmax}$ derived from SIF has captured much of the spatial variability in $V_{cmax}$ due to irrigation and soil properties that are not captured by optimality theory. Remote sensing products can therefore provide more nuanced information on plant responses to non-meteorological environmental drivers and can therefore provide more accurate $V_{cmax}$ estimates and additional information on its spatial variability.


In addition to soil properties, leaf traits are expected to be more directly related to $V_{cmax}$. We use LCC, which contains part of the leaf photosynthetic nitrogen pool (Xu et al., 2012), as an indicator of the effect of leaf traits on $V_{cmax}$. We found that $\Delta V_{cmax}$ is significantly and positively correlated to LCC for individual PFTs ($r^2$=0.0004-0.35, P<0.001 for TROPOMI) and for all PFTs combined ($r^2$=0.25, P<0.001) (Figure 6). Similarly, the relative difference in EOT-derived $V_{cmax}$ and ground

measurements is also significantly correlated with leaf nitrogen content ($r^2$=0.21, P<0.001, Figure 7), in agreement with the finding of Smith et al. (2019). These positive relationships suggest that LCC as a proxy of the photosynthetic nitrogen content in leaves can explain part of the spatial variability in $V_{cmax}$ due to the variations in environmental conditions that are not captured by the optimality theory. The added information from LCC due to plants' optimal allocation of nutrient resources would be useful to improve optimality-based modeling of $V_{cmax}$.


The global distribution patterns of growing season Vcmax shown by the four products (Figure 1) have common latitudinal gradients, i.e. Vcmax is generally largest near the equator and decreases away from the equator. This latitudinal dependence is simulated in EOT through considering radiation, i.e. stronger radiation leading to larger Vcmax. This influence of radiation on Vcmax is well captured by the three remote sensing products. According to analyses of the leaf economics spectrum (Wright

et al., 2004; Sack et al., 2013; Osnas et al., 2013; Reich 2014), leaf photosynthetic capacity increases with mean annual rainfall, and therefore Vcmax in dry areas is expected to be smaller than that in wet areas. However, several semi-arid regions, such as India, middle East, southeast Brazil, and areas near the southern border of the Sahara desert, have large Vcmax values. We found that these are mostly irrigated agricultural areas (Figure 4) and the high Vcmax values there are due to crop management and are not in contradiction to existing leaf economics spectrum data. The Vcmax distribution pattern in Australia is compatible

with the rainfall distribution, i.e. the lowest Vcmax is found in central Australia where rainfall is lowest and the highest Vcmax is located in northern Australia where rainfall is largest. The EOT product can also capture this pattern to some extent through meteorological variables (e.g. radiation and temperature). There are many spatial details in these Vcmax products that are of great interest to leaf economic studies.

### 3.3. Global Mean Values of $V_{cmax}$ for Different Biomes

We compared the mean $V_{cmax}$ values of the TROPOMI SIF+LCC product, denoted by $V_{cmaxTg}$, representing the three remote sensing products, and the EOT product over the growing season grouped by PFT with ground-based datasets at the mean growth temperature and at 25 ℃ after temperature normalization using the same scheme of Smith et al. (2019) (Table 2). The agreement between TROPOMI and EOT is best for EBF, DBF, SHR and GRS, for which the difference between the two

products is smaller than their mean standard deviation. For ENF, DNF and CRP, the difference between the two products exceeded their mean standard deviation. TROPOMI $V_{cmaxTg}$ is also considerably smaller than the ground datasets because of the large contributions of high latitude conifer forests with low $V_{cmaxTg}$ that are underrepresented in the ground datasets (Figure 8). Since the TROPOMI $V_{cmaxTg}$ product compares well with the ground database (Figure 1 and Table 1) and has complete coverage for each PFT, it provides more reliable global averages than the ground database shown in Table 2. Ground data for

DNF are too few (Figure 8) to make sound evaluation for this PFT. TROPOMI $V_{cmaxTg}$ is considerably larger than EOT $V_{cmaxTg}$ for CRP, as well as GRS to a less extent, mostly because of the positive impact of irrigation on $V_{cmaxTg}$ as demonstrated in Figure 4. Although the same temperature function is used in the normalization for all products, the relative changes from $V_{cmaxTg}$ to $V_{cmax25}$ for the various PFTs differed slightly among the four global products (Table 2) as the differences in $V_{cmaxTg}$ among the products vary spatially with different growth temperatures, creating different weights in the calculation of the global

averages of $V_{cmax25}$.

In addition to the TROPOMI $V_{cmax}$ product, the other two remote sensing products are also compared in Figure 9. The magnitude of $V_{cmax25}$ in the TROPOMI product is generally in between those from GOME-2 SIF and LCC for the various PFTs (Figure 9a) because it uses information of LCC which tends to be converted to lower values of $V_{cmax}$ using existing empirical

relationships. For forest PFTs, both $V_{cmax25}$ and $V_{cmaxTg}$ of the EOT product is generally larger than those of remote sensing products. This is likely due to the fact that EOT considers only meteorological variables while soil nutrients and other variables could impose limitations on plant growth and hence leaf traits while remote sensing techniques could be responsive to these soil effects on plants. For the same reason, $V_{cmax25}$ and $V_{cmaxTg}$ values of the EOT product for CRP is smaller than those of

remote sensing products because crop irrigation and soil pH could have positive effects on leaf $V_{cmax}$ that are captured by the
remote sensing products but not by EOT (Figures 3 and 4). The mean values of $V_{cmax25}$ and $V_{cmaxTg}$ from the ground databases
(Smith et al., 1999; Kattge et al., 2009) are given in Figure 9 for comparison purposes, but they don't represent the true global
averages for the various PFTs because of their limited spatial distributions (Figure 8 for Smith et al., 2019). We therefore don't
yet have true ground averages to determine which product provides the most reliable global averages for the various PFTs.
However, based on the point-to-point comparisons (Figure 1 and Table 1), we believe that the TROPOMI product is most
reliable in providing the global averages of $V_{cmax25}$ and $V_{cmaxTg}$ for the various PFTs.

## 4. Discussion

The $V_{cmax}$ datasets derived from SIF and LCC represent the *in situ* leaf-level $V_{cmax}$ that is the collective outcome of
meteorological conditions and other environmental properties. These datasets can therefore be used directly in TEMs without
further adjustment. The TRY database (Kattge et al., 2009) contains both $V_{cmax}$ normalized to 25°C ($V_{cmax25}$) and total LNC,
and they are well correlated. Empirical studies have also shown this correlation (Ryan, 1995; Medlyn et al., 1999; Walker et
al., 2014; Prentice et al., 2014). LNC has therefore been used to adjust $V_{cmax25}$ within the same PFT in some TEMs. However,
such an adjustment can only recover part of the spatio-temporal variability in $V_{cmax25}$ because only a small part of LNC is
closely linked to carboxylation capacity. LNC can be separated into four components: photosynthetic, structural, storage and
respiratory nitrogen pools (Xu et al., 2012; Ali et al., 2016). The photosynthetic nitrogen pool can be further divided into sub-
pools for light harvesting, electronic transport and carboxylation, and its fraction to the total LNC is variable depending on
meteorological and soil conditions and possibly also atmospheric $CO_2$ concentration (Ali et al., 2016). For fully grown leaves
in balance with environmental conditions, these sub-pools are naturally optimized so that the investment of resources in light-
harvesting optimally satisfies the need for electron transport or carboxylation (Xu et al., 2012). In other words, photosynthetic
subpools are highly correlated, giving rise to the experimental evidence that LCC containing the light-harvesting nitrogen is
highly correlated to $V_{cmax}$ and $J_{max}$ (Croft et al., 2017). The daunting and complex task of mapping the spatio-temporal
distributions of leaf photosynthetic capacity could therefore be accomplished by mapping LCC that contains nitrogen in
balance with carboxylation nitrogen in Rubisco, and multispectral or hyperspectral remote sensing data that are highly sensitive
to light absorption by the chlorophyll pigments would be a reliable way to obtain such highly desirable information. The LCC
product shown in this study could therefore be used in conjunction with $V_{cmax}$ products derived from SIF and optimality theory
to parameterize $V_{cmax}$ models with consideration of the nitrogen cycle.

Our remote sensing algorithms derive Vcmax from SIF and LCC from multi-spectral data from sunlit leaves after considering
the Sun-target-view geometry (He et al., 2019; Croft et al., 2020), and hence the remote sensing Vcmax products represent
sunlit leaves observed by the sensors. The observed sunlit leaves are mostly located near the top of the canopy, and hence these
Vcmax products could be considered to represent the average condition of leaves near the top of the canopy. In applying a
Vcmax value to a canopy, it would be necessary to consider the vertical variation of Vcmax in the canopy. A mathematical

scheme to integrate the vertical variation for average sunlit and shaded leaves at different LAI values and solar zenith angles is available from Chen et al. (2012).

The growing season mean Vcmax products are available at https://doi.org/10.5281/zenodo.6466968, but seasonal variation of Vcmax is not yet ready for distribution. Reliable seasonal variation of Vcmax is not yet produced at the global scale due to several reasons: (1) SIF data are often not reliable over non-growing seasons and near the beginning and end of the growing season; (2) LCC derivation is considerably affected by the inaccuracy in the input LAI data outside of the growing season, and near the beginning and end of the growing season the separation of LCC and LAI signals in remote sensing data is still an

issue; (3) the ecological optimality theory that provides the basis for evaluating remote sensing Vcmax products can so far be used for calculating growing season mean Vcmax and is not yet ready for calculating its seasonal variation; and (4) few ground-based data with seasonal variation of Vcmax are available for validation. While efforts are being made to overcome these issues, it will take a while to accumulate sufficient ground-based datasets and to improve remote sensing algorithms and the optimality theory before reliable seasonal variation of Vcmax can be derived at the global scale.


## 5. Conclusion

The two RS $V_{cmax}$ products used in this research were derived independently from separate satellite observations of SIF and LCC, and yet show close agreement in their magnitudes and spatial patterns of modelled Vcmax. These remotely sensed $V_{cmax}$ products(https://doi.org/10.5281/zenodo.6466968) also closely agree in large-scale spatial patterns with those calculated from

the ecological optimality theory using meteorological variables, providing support for the use the theory for prognostic modeling of terrestrial ecosystem function under future climate scenarios. However, the optimality-based $V_{cmax}$ product does not show the local-scale spatial distribution patterns that are consistently found in all three remote sensing products because of patchy land cover distributions, implying that meteorological variables alone do not capture all spatial variability. Importantly, the relative difference in $V_{cmax}$ ($\Delta V_{cmax}$) between SIF and optimality-based products is found to be significantly

correlated to the fraction of irrigation area in a pixel, soil pH and leaf nitrogen content; highlighting the impacts of environmental conditions on $V_{cmax}$ that are not captured within optimality theory. From these results, we conclude: (1) the remote sensing products shown in this study have reliably captured the spatial variability in $V_{cmax}$ and therefore are directly useful for diagnostic ecological modeling at the global scale, and (2) in comparison to the optimality-based product, the remote sensing products provide additional information on how $V_{cmax}$ varies according to local environmental conditions, which is

useful for prognostic modeling purposes. Furthermore, understanding the dynamic in situ response of plant photosynthetic capacity to soil water and nutrient availability, independent of meteorological drivers, is important to monitoring plant photosynthetic potential. The LCC product shown in this study could be used in conjunction with $V_{cmax}$ products derived from SIF and optimality theory to parameterize $V_{cmax}$ models with consideration of the nitrogen cycle. This work demonstrates the

power of global-scale satellite-based and ecological optimality approaches to reveal crucial spatial information on $V_{cmax}$; thereby removing a barrier in global ecological and Earth system research.

**Acknowledgement**

JMC and RW acknowledge support of Chinese National Global Change Key Research Program (2020YFA0608701) and Natural Science and Engineering Council of Canada (RGPIN-2020-05163) for this research. NGS acknowledges support from the National Science Foundation (DEB-2045968). TFK acknowledges support from a NASA Carbon Cycle Science Award 80NSSC21K1705, and the RUBISCO SFA, which is sponsored by the Regional and Global Model Analysis (RGMA) Program in the Climate and Environmental Sciences Division (CESD) of the Office of Biological and Environmental Research (BER) in the U.S. Department of Energy (DOE) Office of Science. TFK, ICP, NS and WH acknowledge the LEMONTREE (Land Ecosystem Models based On New Theory, obseRvations and ExperimEnts) project, funded through the generosity of Eric and Wendy Schmidt by recommendation of the Schmidt Futures programme. ICP acknowledges support from the European Research Council (ERC) under the European Union's Horizon 2020 research and innovation programme (grant agreement No: 787203 REALM) and the LEMONTREE (Land Ecosystem Models based On New Theory, observation and Experiments) project, funded through the generosity of Eric and Wendy Schmidt by recommendation of the Schmidt Futures programme. ND is supported by the European Research Council (ERC) under the European Union's Horizon 2020 research and innovation programme (grant agreement No: 787203 REALM).

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

**Table 1. Correlations by plant function type between $V_{cmax}$ at growing season mean temperature (Tg) in four products (GOME-2 SIF, TROPOMI SIF, LCC and EOT) and a ground database with 3672 individual data points aggregated to 180 grids of 0.5º resolution.**

| Product | $r^2$, p and RMSE ($\mu$ mol m$^{-2}$s$^{-1}$) | ENF, DNF, DBF (n=44) | EBF (n=58) | GRS (n=40) | CRP (n=39) | SHR (n=6) | ALL (n=187) |
|---|---|---|---|---|---|---|---|
| GOME-2 | $r^2$ | 0.15 | 0.35 | 0.83 | 0.27 | 0.32 | 0.69 |
| | RMSE | 8.26 | 15.73 | 17.37 | 11.41 | 13.58 | 13.80 |
| | p | <0.01 | <0.001 | <0.001 | <0.001 | 0.25 | <0.001 |
| TROPOMI | $r^2$ | 0.31 | 0.66 | 0.85 | 0.65 | 0.90 | 0.80 |
| | RMSE | 5.57 | 9.55 | 13.32 | 5.50 | 5.55 | 8.99 |
| | p | <0.001 | <0.001 | <0.001 | <0.001 | <0.01 | <0.001 |
| LCC | $r^2$ | 0.01 | 0.18 | 0.77 | 0.30 | 0.76 | 0.55 |
| | RMSE | 8.4 | 25.47 | 21.06 | 9.14 | 11.35 | 18.28 |
| | p | 0.54 | <0.001 | <0.001 | <0.001 | <0.05 | <0.001 |
| EOT | $r^2$ | 0.10 | 0.34 | 0.85 | 0.38 | 0.22 | 0.66 |
| | RMSE | 7.70 | 19.57 | 11.78 | 7.42 | 12.42 | 13.37 |
| | p | 0.042 | <0.001 | <0.001 | <0.001 | 0.35 | 0.001 |



**Table 2 Mean and standard deviation (SD) of $V_{cmax}$ at the growing temperature ($V_{cmaxTg}$) and normalized to 25 ºC ($V_{cmax25}$) for**
**different plant functional types (PFT) calculated from the TROPOMI and ecological optimality theory (EOT) products in**
**comparison with two ground-based databases (Smith et al., 2019 and Kattge et al., 2009).**

| PFT | ($\mu$mol m$^{-2}$s$^{-1}$) | TROPOMI | | EOT | | Smith 2019 | | Kattge 2009 | |
|---|---|---|---|---|---|---|---|---|---|
| | | Mean | SD | Mean | SD | Mean | SD | Mean | SD |
| ENF | $V_{cmax25}$ | 32.36 | 12.51 | 60.66 | 7.19 | 53.70 | 26.95 | 62.50 | 24.70 |
| | $V_{cmaxTg}$ | 7.31 | 3.62 | 13.68 | 2.97 | 17.43 | 11.13 | | |
| EBF | $V_{cmax25}$ | 46.89 | 13.02 | 54.55 | 6.79 | 45.83 | 23.27 | 43.80 | 16.83 |
| | $V_{cmaxTg}$ | 44.22 | 15.98 | 50.88 | 12.19 | 37.12 | 23.59 | | |
| DNF | $V_{cmax25}$ | 44.38 | 8.93 | 60.50 | 5.05 | 44.82 | 23.34 | 39.10 | 11.70 |
| | $V_{cmaxTg}$ | 10.95 | 2.58 | 14.93 | 2.09 | 11.59 | 6.28 | | |
| DBF | $V_{cmax25}$ | 44.42 | 16.42 | 59.60 | 6.31 | 51.31 | 25.06 | 57.70 | 21.20 |
| | $V_{cmaxTg}$ | 18.12 | 17.07 | 22.68 | 15.68 | 24.31 | 20.72 | | |
| SHR | $V_{cmax25}$ | 53.30 | 13.60 | 61.37 | 7.55 | 50.63 | 27.75 | 57.85 | 19.55 |
| | $V_{cmaxTg}$ | 13.21 | 11.24 | 15.76 | 14.54 | 31.88 | 27.80 | | |
| GRS | $V_{cmax25}$ | 74.74 | 22.76 | 69.45 | 12.37 | 82.70 | 47.86 | 78.20 | 31.10 |
| | $V_{cmaxTg}$ | 49.30 | 40.10 | 41.42 | 27.85 | 21.65 | 18.25 | | |
| CRP | $V_{cmax25}$ | 87.57 | 17.42 | 62.12 | 9.59 | 90.21 | 32.13 | 100.70 | 36.60 |
| | $V_{cmaxTg}$ | 54.83 | 37.14 | 39.63 | 26.72 | 42.11 | 22.64 | | |







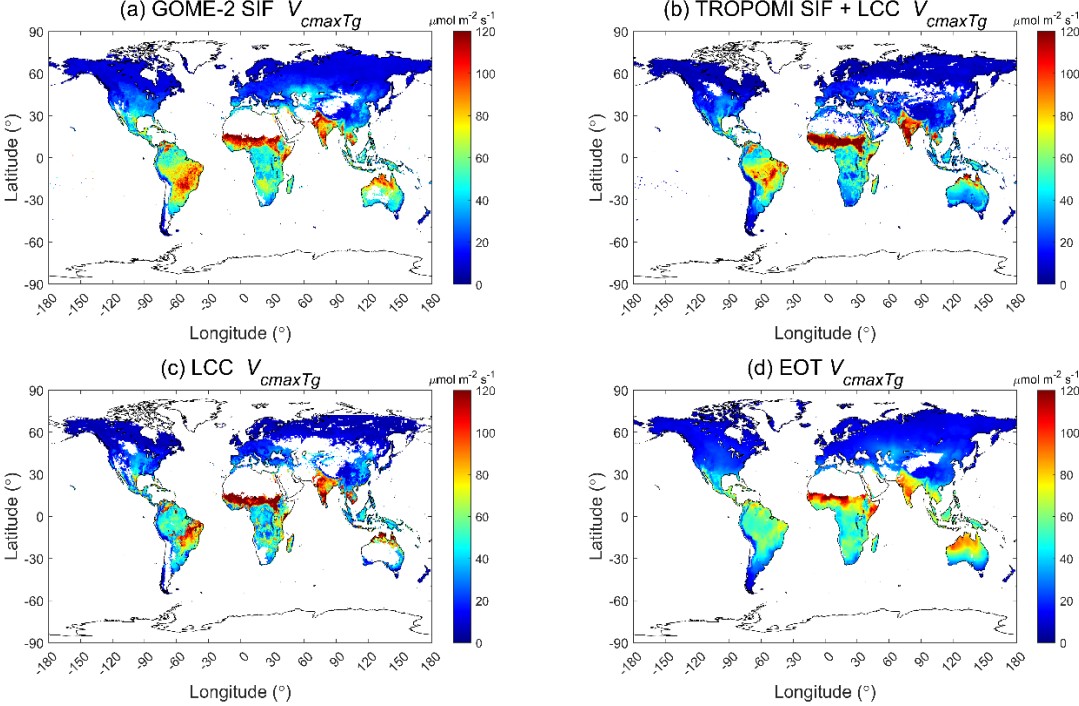


**Figure 1. Global distributions of $V_{\text{cmax}Tg}$ at the mean growing season temperature derived using (a) GOME-2 SIF (2007-2011), (b) TROPOMI SIF+LCC (2019) constrained by leaf chlorophyll content (LCC), (c) LCC (2017), and (d) ecological optimality theory (1901-2015). White areas are missing data**

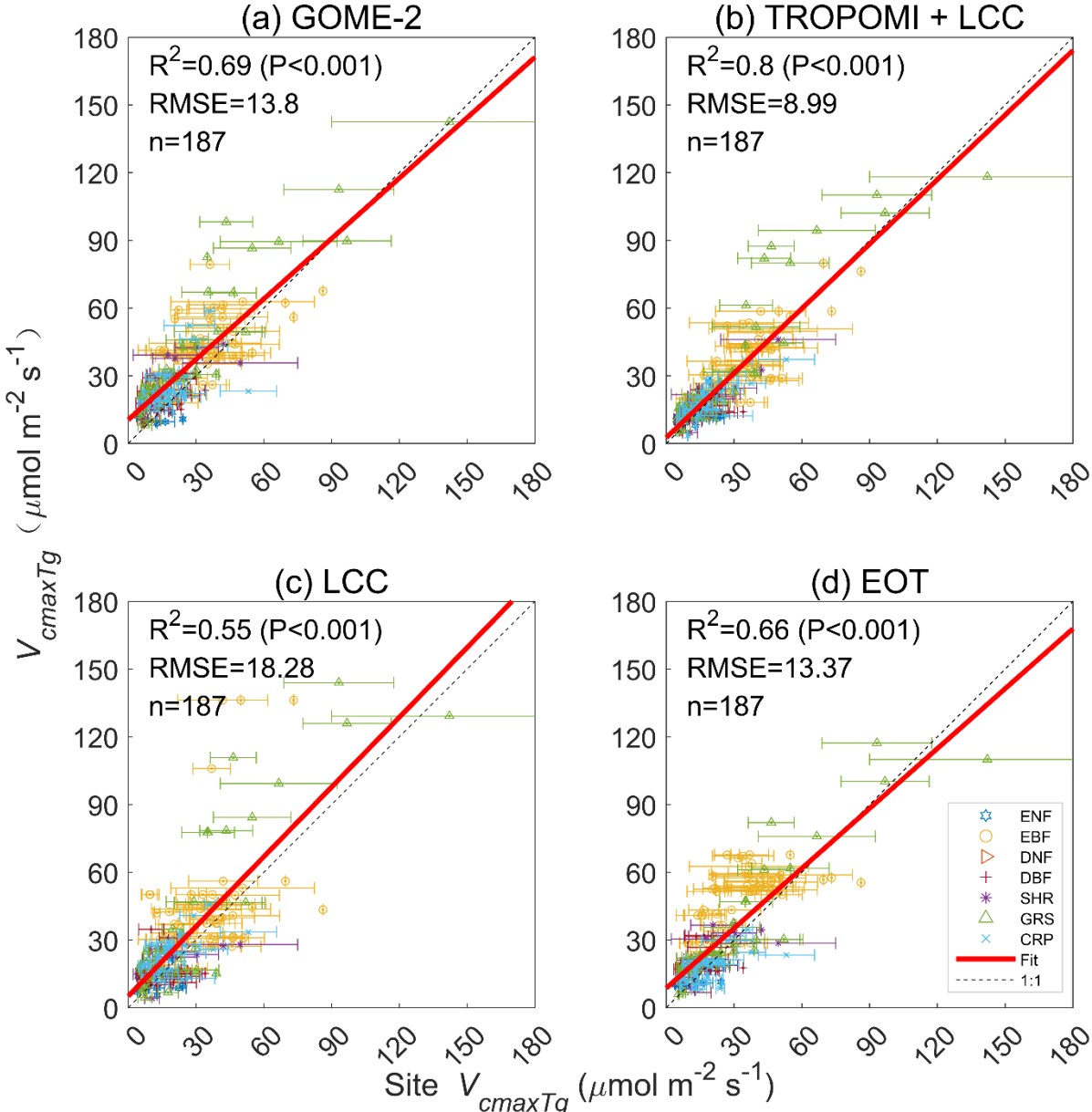

**Figure 2. Comparisons of $V_{cmax}$ at growing season mean temperature (Tg) derived from GOME-2 SIF, TROPOMI SIF+LCC, LCC and optimality theory (EOT) against a ground database with 3672 individual data points aggregated to 180 grids of 0.5° resolution. The root mean square error (RMSE) is in unit of $\mu$ mol m$^{-2}$s$^{-1}$.**

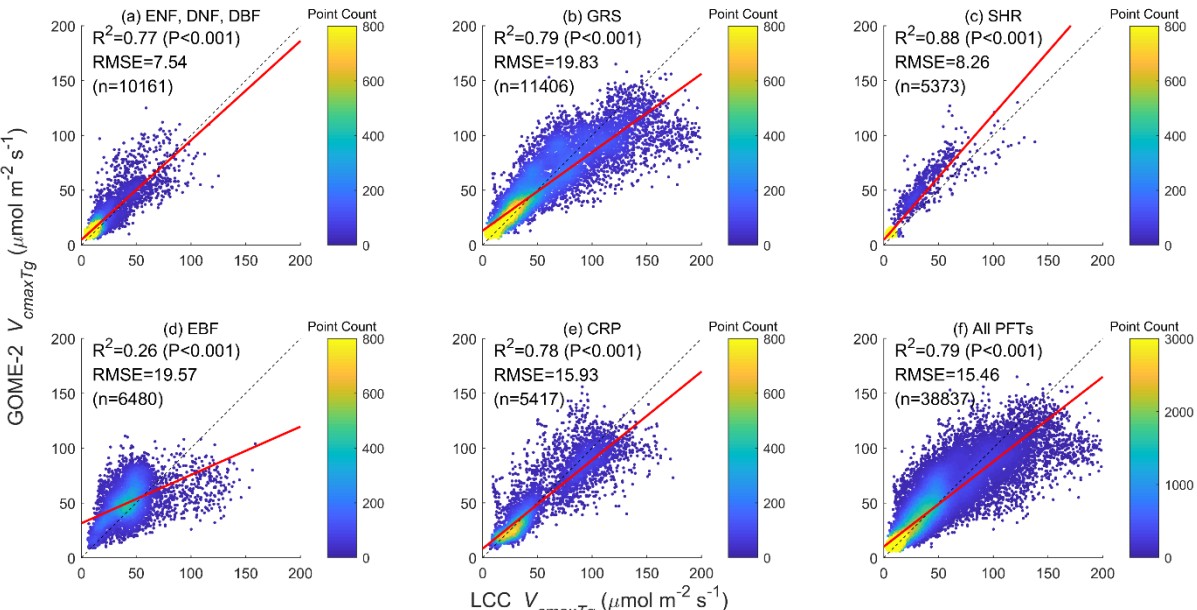

**Figure 3. Comparisons of SIF-derived and LCC-derived $V_{cmax}$ values for a group of three PFTs and four individual PFTs as well as all PFTs combined. These two sets of $V_{cmax}$ derived independently using two different remote sensing techniques are very well correlated for all PFTs except for the evergreen broadleaf forests (EBF) in tropical areas where frequent clouds degrade the quality of both SIF and LCC datasets.**

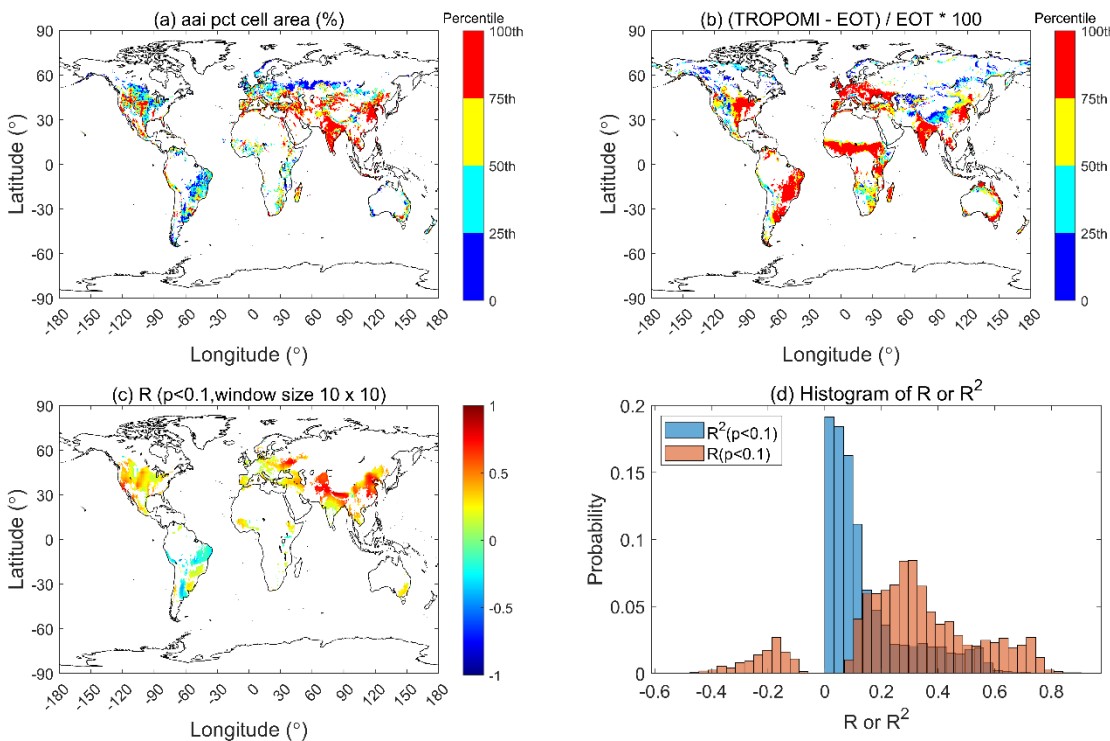


**Figure 4. The influence of irrigation on $V_{cmax}$ over cropland and grassland, detected by TROPOMI SIF+LCC at 0.5° resolution, where (a) is the actual area irrigated in percent of cell area (aai pct cell area) in recent decades, (b) the relative difference in $V_{cmax}$ ($\Delta V_{cmax}$) between TROPOMI and ecological optimality theory (EOT), i.e. $\Delta V_{cmax}$=(TROPOMI-EOT)/EOT, (c) the correlation coefficient (R) between actual irrigated area percentage and $\Delta V_{cmax}$ within sliding windows of 10 x10 pixels, and (d) the histograms of R and $R^2$ values in (c) for cropland and grassland. $\Delta V_{cmax}$ is significantly correlated with percent area irrigated in both cropland (R=0.32, p<0.001) and grassland (R=0.30, p<0.001) at the global scale.**

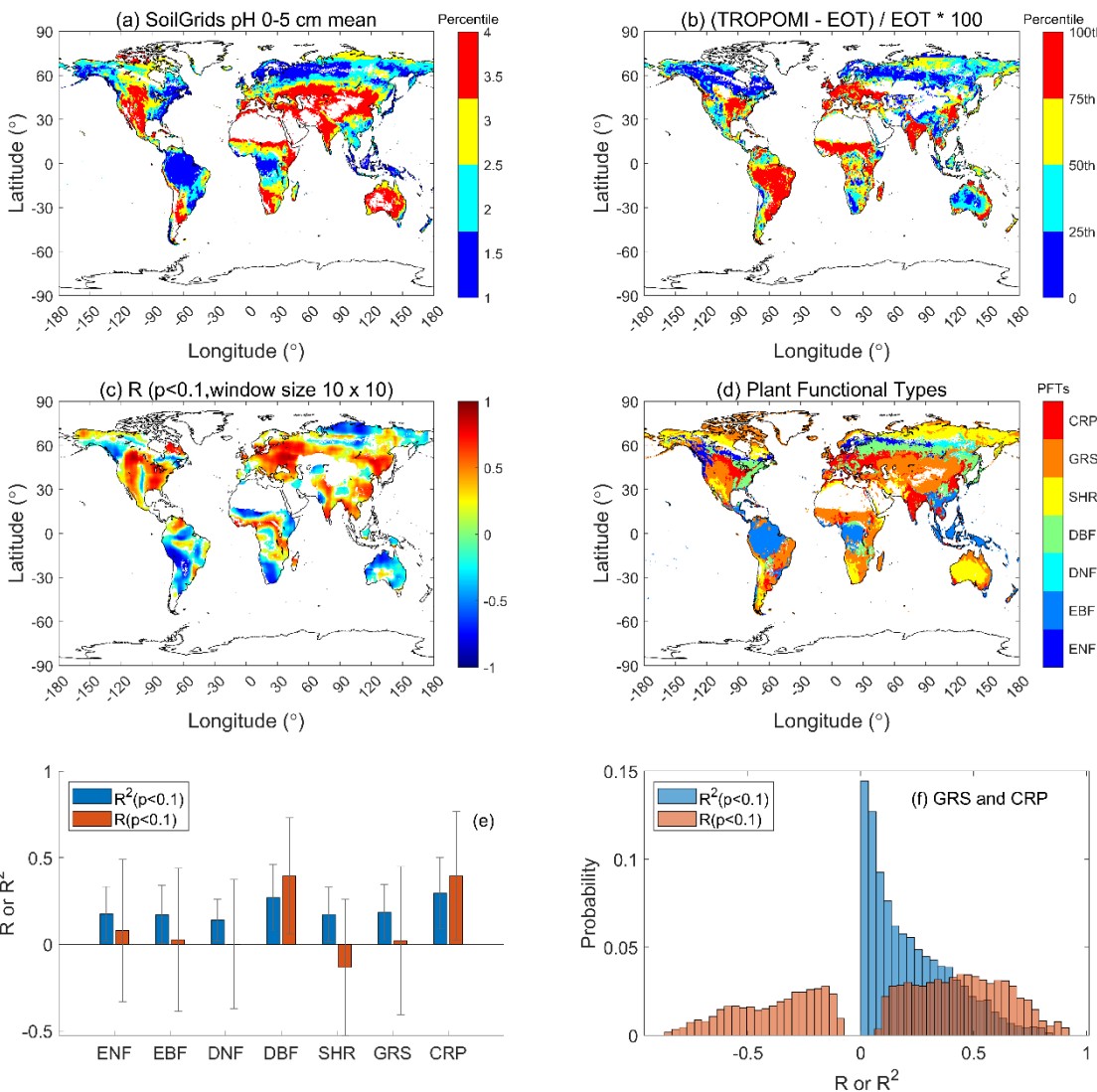


**Figure 5. Soil pH has significant influence on $V_{cmax}$ detected by TROPOMI SIF+LCC at 0.5° resolution. (a) soil pH in the top 0-5 cm layer, (b) relative difference in $V_{cmax}$ ($\Delta V_{cmax}$) between TROPOMI and ecological optimality theory (EOT), i.e. $\Delta V_{cmax}$=(TROPOMI-EOT)/EOT, (c) correlation coefficient (R) between soil pH and $\Delta V_{cmax}$ within sliding windows of 10 x10 pixels, (d) PFT distribution, (e) summary of mean correlation coefficient R and $R^2$ values in (c) by PFTs, and (f) histograms of R and $R^2$**

**values in (c) for grassland (GRS) and cropland (CRP). In 40.3% of GRS and CRP pixels, $\Delta V_{cmax}$ is positively and significantly (p<0.1) correlated with soil pH.**

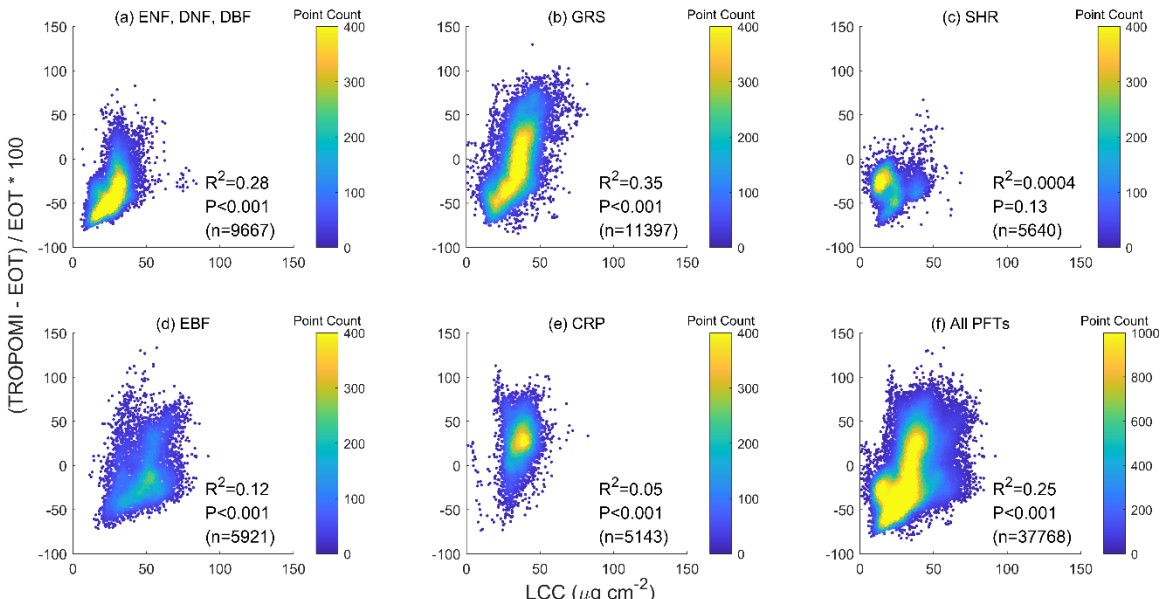


**Figure 6. The relative difference in $V_{cmaxTg}$ ($\Delta V_{cmax}$) between TROPOMI and ecological optimality theory (EOT), i.e. $\Delta V_{cmax}$=(TROPOMI-EOT)/EOT, is significantly correlated to leaf chlorophyll content (LCC) as a proxy of the leaf nutrient condition. All PFTs are included. The correlation is statistically highly significant with p<0.001 for individual PFTs and for all PFTs combined.**


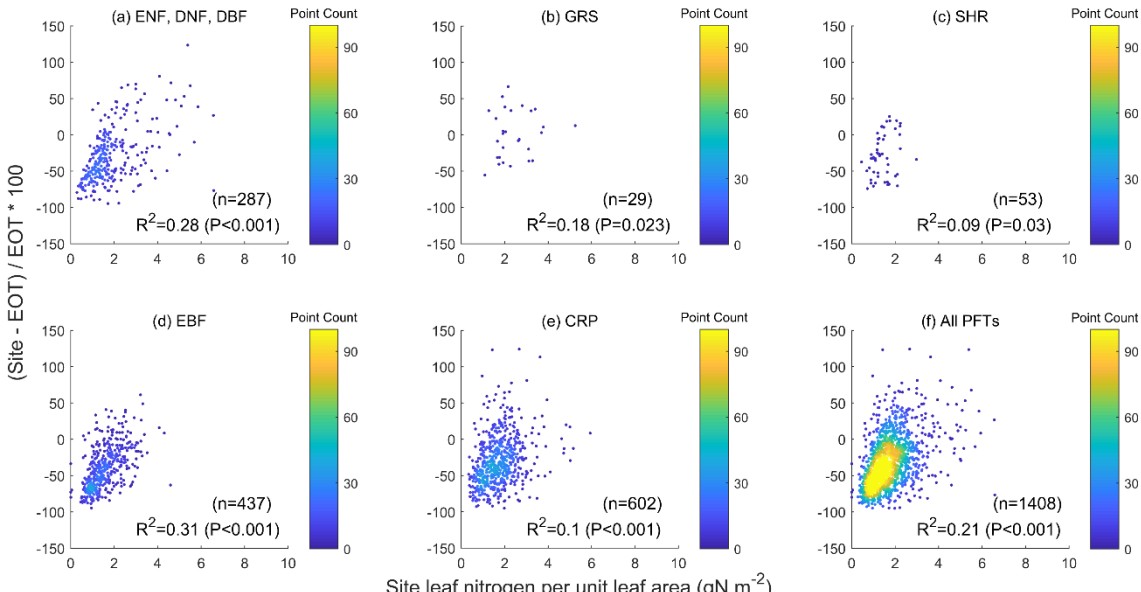


**Figure 7. Influence of leaf nitrogen content on the relative difference between $V_{cmaxTg}$ values measured at ground sites and derived from an ecological optimality theory (EOT) using the available database (Smith et al., 2019). The influence is highly significant for all plant functional types (i.e. p<0.001). The slopes of the regressions of the relative difference in $V_{cmax}$ against LCC or ground leaf nitrogen data are similar, in agreement with the global modeling results that levels of nutrient limitation to plant growth are**

**similar among different PFTs (Fisher et al., 2012).**


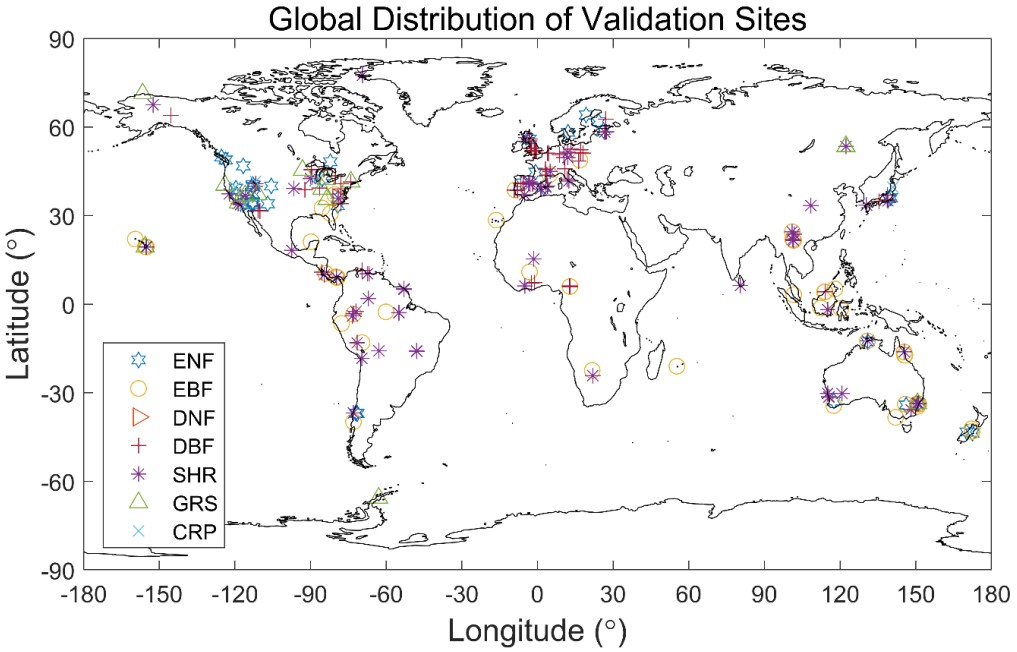

**Figure 8. Distribution of ground sites of the database of Smith et al. (2019) after aggregation to 0.5 grids for the different plant functional types.**

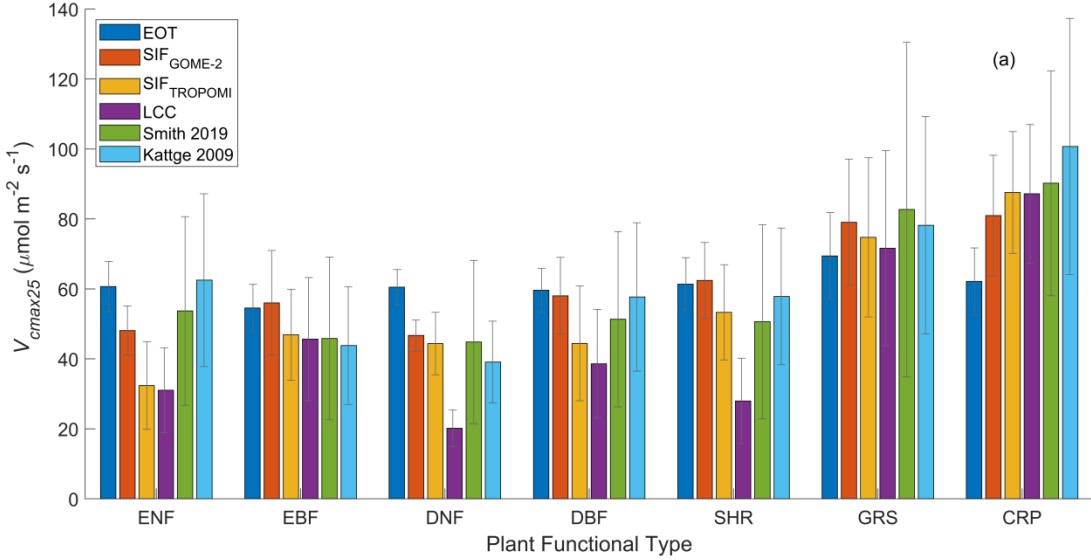

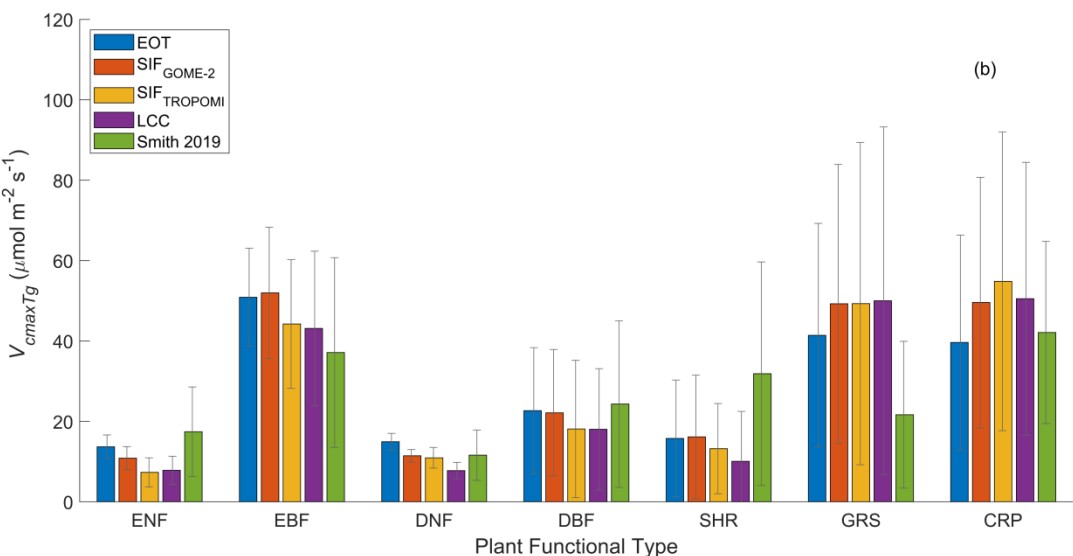

**Figure 9. Mean and standard deviation of $V_{cmaxTg}$ at growth temperature and Vcmax25 (normalized to 25 °C) derived from GOME-2 SIF, TROPOMI SIF+LCC, LCC and ecological optimality theory (EOT) in comparison with two ground databases (Smith 2019 and Kattge 2009) for the main PFTs at growth temperature. Kattge 2009 contains more $V_{cmax25}$ than $V_{cmaxTg}$ so only $V_{cmax25}$ is included in (a). The EOT product has considerably smaller $V_{cmaxTg}$ in grassland (GRS) and crops (CRP) than the three remote sensing products. All four products have considerably higher $V_{cmaxTg}$ than the ground site measurements in grassland mostly because the number of site measurements are too small to be representative of the global average. After the temperature normalization, the differences among the products become much smaller.**

## Code/Data Availability

The $V_{cmax}$ datasets presented in this paper are available at https://doi.org/10.5281/zenodo.6466968 (Chen et al., 2022). It includes the following three global 0.5 degree Vcmax datasets at growth temperature:

1) $V_{cmax}$ from GOME-2 SIF: GOME2_Vcmax_Tg_05deg.tif

2)$V_{cmax}$ from TROPOMI SIF+LCC: TROPOMI_Vmax_Tg_mean.mat

3) $V_{cmax}$ from global leaf chlorophyll content map (Croft et al., 2020, RSE): LCC_Vcmax_Tg_mean.mat

The geographic reference are the same for all three datasets, conforming to that in the geotiff file.

Any questions on the dataset, please contact: Dr. Jing M. Chen, jing.chen@utoronto.ca.

The functions written in R for calculating $V_{cmax}$ using the ecological optimality theory are available at https://github.com/SmithEcophysLab/optimal_vcmax_R (Smith et al., 2022).

## Author Contribution

Conceptualization: JMC, TFK, ICP
Data curation: RW, YL, HC, NGS
Formal analysis: JMC, RW, YL
Funding acquisition: JMC, NS, TK, CP
Investigation: JMC, RW, YL
Methodology: YL, LH, HC, XL, HW
Software: RW, LH, YL
Supervision: JMC
Validation: RW, NGS, TK
Visualization: RW, YL
Writing—original draft: JMC
Writing—review & editing: HC, RW, NGS, TK, ICP, HW, WJ, YZ, ND

## Competing Interests

The authors declare that they have no conflict of interest.

## Right to reproduce any materials

Not applicable