# Peer review of "Global Datasets of Leaf Photosynthetic Capacity for Ecological and Earth System Research"

_Earth System Science Data, 2022_

## Author Comment (AC1)

Title: Global Datasets of Leaf Photosynthetic Capacity for Ecological and Earth System Research

Author(s): Jing M. Chen, Rong Wang, Yihong Liu, Liming He, Holly Croft, Xiangzhong Luo, Han Wang, Nicholas G. Smith, Trevor F. Keenan, I. Colin Prentice, Yongguang Zhang, Weimin Ju, and Ning Dong

MS No.: essd-2022-136

Answer to Reviewer #1: Yao Zhang

Mapping the dynamics of Vcmax at global scale is important for the improvement of the model performance in predicting GPP and to understand the driving factors for its spatial and temporal variations. Recent studies have developed multiple methods to retrieve Vcmax based on satellite observations. This paper by Chen et al. summarized these approaches and provide a direct comparison between these datasets, the one predicted by optimality theory (EOT) as well as in situ observations. The satellite-based datasets generally show good consistency with the EOT and observations. The authors also evaluate the difference between the satellite observations and EOT and suggest that the difference can be explained by irrigation, soil PH, and nitrogen content. This is a solid paper and the developed datasets should be published. However, I still have some comments for the improvement of the manuscript.

Answer: Thank you for your appreciation of our work, and the critical and useful comments below that help improve our manuscript.

In the abstract, the authors mentioned that they use a data assimilation technique to combine the SIF generated Vcmax and LCC generated Vcmax to get an optimized Vcmax, I did not find the description of this data assimilation method. Later in the results, I feel that the authors are referring the TROPOMI SIF based Vcmax as the assimilated Vcmax. If this is the case, the presentation in the abstract should be revised. In the abstract, the authors suggest that the data assimilation technique is to combine "two types" of remote sensing dataset, one is SIF based, the other is LCC based. Clearly, TROPOMI SIF Vcmax, based on its names, should still be considered as SIF based. This naming system is misleading to the readers. I would suggest the authors to reconsider this naming system or revise the abstract.

We agree that the Vcmax product using the combined information of SIF and LCC was not clearly described in Abstract, although it was described in Methods (lines 123-130). We have modified Abstract in response, and TROPOMI SIF Vcmax has been changed to TROPOMI SIF+LCC Vcmax throughout the paper.

The authors suggested that irrigation may be the reason to explain the difference between satellite observed Vcmax and EOT predicted ones. I would argue that the improvement in the crop industry ("green revolution"), mostly better seeds, fertilization usages to be the plausible cause. This is based on the fact that the difference in satellite

and EOT predicted Vcmax is large over all cropland regions, no matter it is irrigated or not (e.g., irrigation cannot explain the difference in Africa and south America). Second, irrigation would provide enough water which tends to reduce Vcmax based on the optimality theory, this is different than what we see in this comparison.

This is an excellent point. The positive effect of irrigation on Vcmax found in our study should be taken as the surrogate effects of agricultural management including not only irrigation but also fertilization and genetic modification. In particular, fertilization usually accompanies with irrigation, as pointed out by Dennis Baldocchi (the other reviewer). Ecological optimality theory predicts higher Vcmax at higher vapor pressure deficit (Smith et al., 2019), which may be related to soil moisture, but the theory has not yet included soil moisture. However, leaf economics spectrum data (Wright et al., 2004; Osnas et al., 2013) show that annual precipitation and Vcmax are positively correlated. We have therefore added the text (Lines 237-240; Lines 266-279) to clarify this issue.

Wright, I. J., P. B. Reich et al., 2004. The worldwide leaf economics spectrum. Nature, 428, 821-827.

Osnas, J. L. D., J. W. Lichstein, P. B. Reich, and S. W. Pacala, 2013. Global leaf trait relationships: mass, area, and the leaf economics spectrum. Science, 340, 741-744.

The manuscript mostly focuses on the comparison of the spatial variation of Vcmax from different datasets. Based on my understanding, all three remote sensing-based Vcamx have seasonal variations. Previous studies have highlighted the importance of correctly representing the seasonal variation of Vcmax to the improvement of seasonal GPP simulations. This seems to be an advantage of the dataset. But I did not see much stress on this temporal variation throughout the manuscript, this is also no cross comparison of these datasets at temporal scales.

Indeed, in this paper we focus on the analysis of spatial variation of Vcmax without exploring its seasonal variation. There are several reasons for this focus: (1) ground-based data used in this study do not have seasonal variation, although there are a limited number of data points with seasonal variation but they are insufficient for validation purposes; (2) the ecological optimality theory can so far be used to derive the mean Vcmax values over the growing season; (3) SIF data are often not reliable over non-growing seasons; and (4) annual patterns of retrieved LCC have irregularities in some places because of inaccuracies of input LAI outside of the growing season. The current state of the art in remote sensing retrievals of Vcmax using SIF and LCC provides reliable values of seasonal maximum and mean of Vcmax, which are a solid step forward, while efforts are being made to retrieve the annual variation of Vcmax through improving the algorithms and developing new algorithms. We expect that Vcmax datasets with reliable annual variations will soon be available. We have added statements (Lines 340-350) on this issue.

Detailed comments:

L31, why three? LCC, SIF and the optimized one?

Yes, we have modified the abstract to make it clear.

L32, the link provides two SIF based Vcmax, which is not described here.

The two SIF-based products are in fact one SIF and one SIF+LCC. We have modified the description on the link.

L48, it would be good to briefly describe how Vcmax can be derived from SIF, you did this for LCC later but not here.

This line is expanded to provide the first principle of deriving Vcmax from SIF (now Lines 50-51).

L64, and SIF is quite noisy.

SIF signals are indeed small and often noisy from various sources including variations in solar observation and sensor view angles. In order not to interrupt the flow of the text, we added a sentence (Lines 101-104) to explain this.

L69, … to produce a global Vcmax time series dataset? Single time series may refer to only one vector.

We agree and have changed "time series" to "map series", and the following sentence is also adjusted to make it consistent.

L98, the SIF-photosynthesis relationship is only linear at longer time scales (weekly or monthly), you may want to specify this. This sentence can be misleading considering you use "instantaneous".

We agree and have changed "is approximately proportional to" to "increases with" in the sentence (now Line 100). The nonlinearity issue is actually considered in our improved Vcmax optimization method (see Lines 125-128).

L100, "sunlit leaves are the predominant sources of SIF" a reference would be helpful here.

A reference of Pinto et al. (2016) is now provided (Line 104).

L150, were these obtained from sunlit leaves only? The remote sensing datasets are for the sunlit leaves, right?

The ground measurements of Vcmax can be made on any leaves, while remote sensing of SIF signals is mostly obtained from sunlit leaves at the time of measurements. However, all leaves in a canopy have probabilities to be sunlit. There is indeed some

mismatch between remote sensing and ground data in terms of leaf sample distribution in the vertical direction in the canopy. This issue deserves further investigation with detailed ground measurements. In response to your and another reviewer's question, a paragraph (Lines 333-339) is added in Discussion regarding the use of Vcmax products for both sunlit and shaded leaves.

L165, I was expected to see the equation here.

As this equation involves many variables and constants, we feel that it is not necessary to take up much space here, but a clear source of the equation is given.

L224, also plant genetic engineering. I think this may be a more plausible reason to explain the difference between TROPOMI and EOT. Human selections are producing much more productive crops that the optimality theory cannot predict. It happens that the much of croplands have irrigation. In Fig. 4b, the different is obvious in almost all croplands across the globe.

While we are not sure if genetic engineering for productive crop species would result in higher leaf Vcmax, we certainly agree that cropland and grassland management through irrigation and fertilization would increase Vcmax. While the impact of irrigation on Vcmax is uncertain, fertilization would directly increase leaf nitrogen and hence Vcmax. Since irrigation and fertilization generally co-occur in cropland and grassland and an irrigation dataset is available at the global scale, we used the irrigation dataset as a surrogate for the cropland and grassland management. We have added a few sentences (Lines 237-240) to clarify the confusion.

L228, but the optimality theory predicts lower Vcmax at regions with abundant water resource.

The optimality theory so far has not considered the influence of soil water on Vcmax, but the water effect may be indirectly considered through air water vapor deficit. Again, the positive irrigation effect on Vcmax stated here is associated with overall cropland and grassland management as explained above.

L258, I think you mean biome level Vcmax here.

Yes, we present global mean values of Vcmax for different biomes in this section. "for different biomes" is added to the section heading.

L259, not sure if TROPOMI is the dataset obtained from data assimilation. This needs to be clarified in the method.

The TROPOMI Vcmax dataset mentioned here is obtained through data assimilation using TROMPOMI SIF data and LCC data described in Methods. To avoid confusion, we

now changed "TROPOMI dataset" to "TROPOMI SIF+LCC dataset", and the Methods section is also slightly modified to reflect the new naming convention (Line 128).

---

## Author Comment (AC2)

Title: Global Datasets of Leaf Photosynthetic Capacity for Ecological and Earth System Research

Author(s): Jing M. Chen, Rong Wang, Yihong Liu, Liming He, Holly Croft, Xiangzhong Luo, Han Wang, Nicholas G. Smith, Trevor F. Keenan, I. Colin Prentice, Yongguang Zhang, Weimin Ju, and Ning Dong

MS No.: essd-2022-136

Answer to Reviewer #2: Dennis Baldocchi

If we accept the Farquhar-von Caemmerer-Berry photosynthetic model as the dominant paradigm for computing leaf and ecosystem photosynthesis and to apply it to the challenge of assessing photosynthesis everywhere and all the time, we will need to assess such key parameters as Vcmax, at a reference temperature. Chen and colleagues have been leading the way in developing a means to do this and here is their global dataset. It profits from the sharing of data by many through the TRY plant traits dataset (>3700 datasets) and use of optimization theory by many of the coauthors and inferences with information from satellite remote sensing to upscale information.

Answer: Thank you for pointing out the merits of this paper and the conditions that made this work possible. These Vcmax datasets presented in this paper are obtained through in-depth collaboration with a large number of individuals in different institutions who contributed in various ways. It would have not been possible to produce and validate these datasets without this collaboration. Vcmax is an indispensable parameter for the FvCB model. Although there are on-going attempts to modify or even replace the model, Vcmax that sets the limit on the carboxylation rate in the dark reaction would still be an essential parameter in any new photosynthetic models.

For the upscaling the authors use two multiple constraints and plausible means, SIF and leaf chlorophyll information deduced from plant reflected spectra. These are useful and defensible. Though I do worry about SIF as the signal is small and many show that it represents absorbed light more. But I don't see this as a fatal flaw and it is worth exploring.

Dechant, B., et al. (2020). "Canopy structure explains the relationship between photosynthesis and sun-induced chlorophyll fluorescence in crops." Remote Sensing of Environment **241**: 111733.

Your concern on the SIF signal is valid, similar to the comment of the other reviewer (Yao Zhang). SIF signals are about 1-5% of the reflected radiation in the near infrared wavelengths. However, at the Fraunhofer lines the signals are much stronger because the reflected radiation is much reduced, giving rise to accurate retrieval of SIF from satellite spectral measurements with high spectral resolutions (better than 0.2 micrometers). We therefore believe that satellite SIF measurements are reliable for the purpose of assessing canopy photosynthesis (many studies showed this) and Vcmax (our studies starting from He et al., 2019). In response to your and Zhang's concern, we have modified some lines (Lines101-104) in Methods and cited Dechant et al., 2020) for angular effects on SIF.

My other words of wisdom, having spent time with books on the ground assessing Vcmax is that we know there is lots of seasonality in this parameter, with changes in leaf allocation of N and effects of soil moisture deficits. But this request may be beyond the scope of this work. But I strongly argue for future efforts to create seasonal maps of Vcmax. My other experience is to find vertical variations in Vcmax with depth in deciduous forests, as there is much light acclimation and strong vertical gradients in leaf N that affect Vcmax. This complication, too, is beyond the scope of this work.

Our current study leads to the production of reliable growing season mean Vcmax maps. Although our remote sensing algorithms allow production of Vcmax maps series with seasonal variation, they are not yet ready for distribution for the following reasons: (1) ground-based data used in this study do not have seasonal variation, although there are a limited number of data points with seasonal variation but they are insufficient for validation purposes; (2) the ecological optimality theory can so far be used to derive the mean Vcmax values over the growing season; (3) SIF data are often not reliable over non-growing seasons; and (4) annual patterns of retrieved LCC have irregularities in some places because of inaccuracies of input LAI. In fact, some progress has been made recently to resolve all these issues, and it is possible to produce a multi-decadal times series of Vcmax with seasonal variations in the near future. We have added a paragraph in Discussion to address this concern (Lines 340-350).

However, the growing season mean Vcmax products are already a large step forward from the current state of the art of using PTF specific constants for Vcmax. The remote sensing products represent the average values of leaves at the top of the canopy, and to obtain the canopy mean Vcmax for sunlit and shaded leaves, we have already developed a vertical integration scheme to consider the vertical gradient of leaf nitrogen content (Chen et al., 2012, Global Biogeochemical Cycles). A paragraph is added in Discussion to ease this common concern (Lines 333-339).

In the methods, I am glad to see the authors consider clumping and sun and shade leaves. This is an effort I would insist upon if one is working on a specific canopy. Though for global assessments I worry that by doing so it may introduce error in Vcmax as we may not now these other factors with enough precision.

This is also a valid concern. In our remote sensing algorithms, we used a global clumping index map at 500 m resolution derived from MODIS data (He et al., 2012) to aid the separation of sunlit and shaded leaves. In our assessment of the CI product against ground data, the error is less than +/-0.1 while the mean values of conifer and broadleaf forests are about 0.53 and 0.66, respectively. Since these mean values are much smaller than unity (the random case), the signal from the CI product is 3-4 times larger than the noise, suggesting that it is highly worthwhile to use the product. Otherwise, the estimation of sunlit and shaded leaf fractions would be in much error, cascading it to Vcmax derivation.

With regards to inverting information derived from leaf chlorophyll I am satisfied to see them using a state of art radiative transfer model, PROSPECT, for this inversion. It is the best way to proceed in my mind. Yes, one may use simple empirical algorithms instead, but are they good enough? Nor may they be mechanistic enough.

This is an excellent insight. In the leaf-level inversion implemented on remote sensing images, we in fact used PTF-specific empirical relationships between LCC and one of two vegetation indices using MERIS red edge bands. The relationships were simulated using the PROSPECT model, in order to attain the necessary computational efficiency (Croft et al., 2020). Considering the differences in input parameters to PROSPECT among PTFs, it was necessary to make the empirical relationships applicable at the global scale.

As noted above using 3700 datasets on A/Ci brings the remote sensing inversion to reality. Can't ask for a better way to do this.

We are fortunate to have this dataset compiled by previous scientists for validation of these new Vcmax products.

Temperature normalization is always the trickiest as we see lots of temperature acclimation in the field. But don't know what else to suggest. Better than nothing.

Since we have only conducted a short period of Vcmax inversion, temperature acclimation is not considered in our temperature normalization. The ecological optimality theory may be adjusted for this purpose if a long-term time series of Vcmax is produced in the future.

Results

While it is nice to see computations compared with ground based measurements, realize that the model is fitted with information from the ground. So a bit circular. Would be better to reserve a subset of data for model testing.   It probably wont change things because with 3700 data points there is over sampling, especially given the scaling work of Reich and others showing that 80% of variances in leaf photosynthesis scales with only a few factors, leaf N, specific leaf weight and age. Maybe comparing your results to this economic leaf scaling result may be a reasonable alternative.

The Vcmax values derived from SIF and LCC are totally independent of the ground-based data used for validation, so it is unnecessary to separate the dataset into training and validation subsets. Scaling Vcmax against other leaf traits (N, SLA, age) is a good idea and would help interpret and evaluate the remote sensing products. There are global leaf economics spectrum datasets used by various studies (Wright et al., 2004; Sack et al., 2013; Osnas et al., 2013; Reich 2014). However, these datasets are collected over a long period of time with different techniques and often without sufficient details of geographical locations for temporal and spatial matching with our remote sensing data over a short period of time. It is possible to do this leaf economic scaling study partially

with our dataset at hand, but we feel that this is an issue for exploring the usefulness of the dataset while the main purpose of this present paper is to show the derivation and information content of this dataset. We have added a paragraph to discuss the possible use of our Vcmax products for leaf economic studies (Lines 266-279).

Wright, I. J., P. B. Reich et al., 2004. The worldwide leaf economics spectrum. Nature, 428, 821-827.

Osnas, J. L. D., J. W. Lichstein, P. B. Reich, and S. W. Pacala, 2013. Global leaf trait relationships: mass, area, and the leaf economics spectrum. Science, 340, 741-744.

Sack, L., C. Scoffoni et al., 2013. How do leaf veins influence the worldwide leaf economic spectrum? Review and synthesis. Journal of Experimental Botany, 64: 4053-4080.

Reich, P. B., 2014, The world-wide "fast-slow" plant economics spectrum: a traits manifesto. Journal of Ecology,102: 275-301.

Glad to see a section on response to drivers. Useful. The issue on irrigation is interesting and could be a scale emergent property from this work. Remember irrigated fields are also fertilized so they will stand out compared to native vegetation.

We agree. Irrigation was used as a surrogate of cropland and grassland management. A limited number of studies showed that crop and grassland water stress decreased Vcmax (Reed and Loik, 2016; Chen et al., 2019; Song et al., 2021). Leaf economics spectrum datasets also show that for natural ecosystems, leaf photosynthetic capacity increases with mean annual precipitation (Wright et al., 2004; Osnas et al., 2013), suggesting that increased water availability in grassland would increase its leaf Vcmax. These could explain partly the positive correlation between Vcmax and irrigation for crops and grassland found in this study. These positive effects could also be associated to fertilization that often co-occurs with irrigation. We have added some discussion on this issue (Lines 237-240).

Chen, B., J. M. Chen, D. D. Baldocchi, Y. Liu, T. Zheng, T. A. Black, and H. Croft. 2019. A new way to include soil water stress in terrestrial ecosystem models. Agricultural and Forest Meteorology, 276, 107649, https://doi.org/10.1016/j.agrformet.2019.107649.

Reed, C. C. and M. E. Loik, 2016. Water relations and photosynthesis along an elevation gradient for Artemisia tridentata during and historic drought. Oecologia, doi: 10.1007/s00442-015-3528-7.

Song X., G. Zhou, Q. He, and H. Zhou, 2021. Quantitative response of maize Vcmax25 to persistent drought stress at different growth stages. Water, 13, https://doi.org/10.3390/w13141971.

Discussion

Looking at your maps I see high Vcmax in desert and semiarid areas (Africa, India, Australia and the Cerrado of Brazil). In my early work on stress, I looked a lot at Park Nobel's work on desert species and indeed did see among the higher Vcmax values. Thinking about Prentice optimization theory I think it makes sense. They need to acquire enough carbon to outpace respiration. But they have a short growing season due to low water supply and high demand. The only way they can make the economics work is to achieve very high rates of photosynthesis, which comes at the cost of high Vcmax and N. I find this interesting and the authors may want to discuss this a bit.

This is an excellent observation and useful suggestion. From the leaf economics perspective, your point of the cost of high Vcmax and N for short growing seasons seems logical. However, the available leaf economics spectrum data (Wright et al., 2004; Osnas et al., 2013) all show that leaf mass per area (LMA) decreases with mean annual rainfall (MAR), leading to higher photosynthetic capacity. It means Vcmax increases with MAR or is lower at drier places. This seems to be opposite to what you expected. The higher Vcmax values in India and southeast Brazil are mostly located in agricultural areas where irrigation might have positive influence on Vcmax. In the areas near the southern border of the Sahara desert, the high Vcmax area is also mostly associated with cropland, and the latitudinal radiation gradient may explain the Vcmax north-south gradient. In Australia, the Vcmax spatial pattern is compatible with precipitation distribution, i.e. low Vcmax in central Australia is associated with low precipitation, while higher Vcmax values in northern Savanah areas are related to higher precipitation. The latitudinal gradient of radiation also enhances the north-south gradient of Vcmax. These Vcmax maps provide a lot of new information for leaf economics studies. We have added a paragraph to discuss the Vcmax distribution patterns in these regions (Lines 266-279).

The quality of the figures is good enough. Looks like they are generated by Matlab and have nice color gradients.

These figures are indeed generated in Matlab!